# Learning to Adapt in Dynamic, Real-World Environments through Meta-Reinforcement Learning

**Anusha Nagabandi\*, Ignasi Clavera\*, Simin Liu,**
**Ronald S. Fearing, Pieter Abbeel, Sergey Levine, & Chelsea Finn**
University of California, Berkeley
{nagaban2,iclavera,simin.liu}@berkeley.edu
{ronf,pabbeel,svlevine,cbfinn}@berkeley.edu

## Abstract

Although reinforcement learning methods can achieve impressive results in simulation, the real world presents two major challenges: generating samples is exceedingly expensive, and unexpected perturbations or unseen situations cause proficient but specialized policies to fail at test time. Given that it is impractical to train separate policies to accommodate all situations the agent may see in the real world, this work proposes to learn how to quickly and effectively adapt online to new tasks. To enable sample-efficient learning, we consider learning online adaptation in the context of model-based reinforcement learning. Our approach uses meta-learning to train a dynamics model prior such that, when combined with recent data, this prior can be rapidly adapted to the local context. Our experiments demonstrate online adaptation for continuous control tasks on both simulated and real-world agents. We first show simulated agents adapting their behavior online to novel terrains, crippled body parts, and highly-dynamic environments. We also illustrate the importance of incorporating online adaptation into autonomous agents that operate in the real world by applying our method to a real dynamic legged millirobot. We demonstrate the agent's learned ability to quickly adapt online to a missing leg, adjust to novel terrains and slopes, account for miscalibration or errors in pose estimation, and compensate for pulling payloads.[1]

## 1 Introduction

Both model-based and model-free reinforcement learning (RL) methods generally operate in one of two regimes: all training is performed in advance, producing a model or policy that can be used at test-time to make decisions in settings that approximately match those seen during training; or, training is performed online (e.g., as in the case of online temporal-difference learning), in which case the agent can slowly modify its behavior as it interacts with the environment. However, in both of these cases, dynamic changes such as failure of a robot's components, encountering a new terrain, environmental factors such as lighting and wind, or other unexpected perturbations, can cause the agent to fail. In contrast, humans can rapidly adapt their behavior to unseen physical perturbations and changes in their dynamics (Braun et al., 2009): adults can learn to walk on crutches in just a few seconds, people can adapt almost instantaneously to picking up an object that is unexpectedly heavy, and children that can walk on carpet and grass can quickly figure out how to walk on ice without having to relearn how to walk. How is this possible? If an agent has encountered a large number of perturbations in the past, it can in principle use that experience to *learn how to adapt*. In this work, we propose a meta-learning approach for learning online adaptation.

Motivated by the ability to tackle real-world applications, we specifically develop a model-based meta-reinforcement learning algorithm. In this setting, data for updating the model is readily available at every timestep in the form of recent experiences. But more crucially, the meta-training process for training such an adaptive model can be much more sample efficient than model-free meta-RL approaches (Duan et al., 2016; Wang et al., 2016; Finn et al., 2017). Further, our approach foregoes

---

[1]Videos available at: https://sites.google.com/berkeley.edu/metaadaptivecontrol

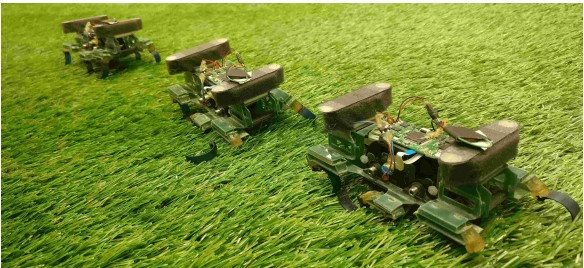

Figure 1: We implement our sample-efficient meta-reinforcement learning algorithm on a real legged millirobot, enabling online adaptation to new tasks and unexpected occurrences such as losing a leg (shown here), novel terrains and slopes, errors in pose estimation, and pulling payloads.

the episodic framework on which model-free meta-RL approaches rely on, where tasks are pre-defined to be different rewards or environments, and tasks exist at the trajectory level only. Instead, our method considers each timestep to potentially be a new "task, " where any detail or setting could have changed at any timestep. This view induces a more general meta-RL problem setting by allowing the notion of a task to represent anything from existing in a different part of the state space, to experiencing disturbances, or attempting to achieve a new goal.

Learning to adapt a model alleviates a central challenge of model-based reinforcement learning: the problem of acquiring a global model that is accurate throughout the entire state space. Furthermore, even if it were practical to train a globally accurate dynamics model, the dynamics inherently change as a function of uncontrollable and often unobservable environmental factors, such as those mentioned above. If we have a model that can adapt online, it need not be perfect everywhere a priori. This property has previously been exploited by adaptive control methods (Åström and Wittenmark, 2013; Sastry and Isidori, 1989; Pastor et al., 2011; Meier et al., 2016); but, scaling such methods to complex tasks and nonlinear systems is exceptionally difficult. Even when working with deep neural networks, which have been used to model complex nonlinear systems (Kurutach et al., 2018), it is exceptionally difficult to enable adaptation, since such models typically require large amounts of data and many gradient steps to learn effectively. By specifically training a neural network model to require only a small amount of experience to adapt, we can enable effective online adaptation in complex environments while putting less pressure on needing a perfect global model.

The primary contribution of our work is an efficient meta reinforcement learning approach that achieves online adaptation in dynamic environments. To the best knowledge of the authors, this is the first meta-reinforcement learning algorithm to be applied in a real robotic system. Our algorithm efficiently trains a global model that is capable to use its recent experiences to quickly adapt, achieving fast online adaptation in dynamic environments. We evaluate two versions of our approach, recurrence-based adaptive learner (ReBAL) and gradient-based adaptive learner (GrBAL) on stochastic and simulated continuous control tasks with complex contact dynamics (Fig. 2). In our experiments, we show a quadrupedal "ant" adapting to the failure of different legs, as well as a "half-cheetah" robot adapting to the failure off different joints, navigating terrains with different slopes, and walking on floating platforms of varying buoyancy. Our model-based meta RL method attains substantial improvement over prior approaches, including standard model-based methods, online model-adaptive methods, model-free methods, and prior meta-reinforcement learning methods, when trained with similar amounts of data. In all experiments, meta-training across multiple tasks is sample efficient, using only the equivalent of $1.5 - 3$ hours of real-world experience, roughly $10\times$ less than what model-free methods require to learn a single task. Finally, we demonstrate GrBAL on a real dynamic legged millirobot (see Fig 2). To highlight not only the sample efficiency of our meta model-based reinforcement learning approach, but also the importance of fast online adaptation in the real world, we show the agent's learned ability to adapt online to tasks such as a missing leg, novel terrains and slopes, miscalibration or errors in pose estimation, and new payloads to be pulled.

## 2 RELATED WORK

Advances in learning control policies have shown success on numerous complex and high dimensional tasks (Schulman et al., 2015; Lillicrap et al., 2015; Mnih et al., 2015; Levine et al., 2016; Silver et al., 2017). While reinforcement learning algorithms provide a framework for learning new tasks, they primarily focus on mastery of individual skills, rather than generalizing and quickly adapting to new scenarios. Furthermore, model-free approaches (Peters and Schaal, 2008) require large amounts of

system interaction to learn successful control policies, which often makes them impractical for real-world systems. In contrast, model-based methods attain superior sample efficiency by first learning a model of system dynamics, and then using that model to optimize a policy (Deisenroth et al., 2013; Lenz et al., 2015; Levine et al., 2016; Nagabandi et al., 2017b; Williams et al., 2017). Our approach alleviates the need to learn a single global model by allowing the model to be adapted automatically to different scenarios online based on recent observations. A key challenge with model-based RL approaches is the difficulty of learning a global model that is accurate for the entire state space. Prior model-based approaches tackled this problem by incorporating model uncertainty using Gaussian Processes (GPs) (Ko and Fox, 2009; Deisenroth and Rasmussen, 2011; Doerr et al., 2017). However, these methods make additional assumptions on the system (such as smoothness), and does not scale to high dimensional environments. Chua et al. (2018) has recently showed that neural networks models can also benefit from incorporating uncertainty, and it can lead to model-based methods that attain model-free performance with a significant reduction on sample complexity. Our approach is orthogonal to theirs, and can benefit from incorporating such uncertainty.

Prior online adaptation approaches (Tanaskovic et al., 2013; Aswani et al., 2012) have aimed to learn an approximate global model and then adapt it at test time. Dynamic evaluation algorithms (Rei, 2015; Krause et al., 2017; 2016; Fortunato et al., 2017), for example, learn an approximate global distribution at training time and adapt those model parameters at test time to fit the current local distribution via gradient descent. There exists extensive prior work on online adaptation in model-based reinforcement learning and adaptive control (Sastry and Isidori, 1989). In contrast from inverse model adaptation (Kelouwani et al., 2012; Underwood and Husain, 2010; Pastor et al., 2011; Meier et al., 2016; Meier and Schaal, 2016; Rai et al., 2017), we are concerned in the problem of adapting the forward model, closely related to online system identification (Manganiello et al., 2014). Work in model adaptation (Levine and Koltun, 2013; Gu et al., 2016; Fu et al., 2015; Weinstein and Botvinick, 2017) has shown that a perfect global model is not necessary, and prior knowledge can be fine-tuned to handle small changes. These methods, however, face a mismatch between what the model is trained for and how it is used at test time. In this paper, we bridge this gap by explicitly training a model for fast and effective adaptation. As a result, our model achieves more effective adaptation compared to these prior works, as validated in our experiments.

Our problem setting relates to meta-learning, a long-standing problem of interest in machine learning that is concerned with enabling artificial agents to efficiently learn new tasks by learning to learn (Thrun and Pratt, 1998; Schmidhuber and Huber, 1991; Naik and Mammone, 1992; Lake et al., 2015). A meta-learner can control learning through approaches such as deciding the learner's architecture (Baker et al., 2016), or by prescribing an optimization algorithm or update rule for the learner (Bengio et al., 1990; Schmidhuber, 1992; Younger et al., 2001; Andrychowicz et al., 2016; Li and Malik, 2016; Ravi and Larochelle, 2018). Another popular meta-learning approach involves simply unrolling a recurrent neural network (RNN) that ingests the data (Santoro et al., 2016; Munkhdalai and Yu, 2017; Munkhdalai et al., 2017; Mishra et al., 2017) and learns internal representations of the algorithms themselves, one instantiation of our approach (ReBAL) builds on top of these methods. On the other hand, the other instantiation of our method (GrBAL) builds on top of MAML (Finn et al., 2017). GrBAL differs from the supervised version of MAML in that MAML assumes access to a hand-designed distribution of tasks. Instead, one of our primary contributions is the online formulation of meta-learning, where tasks correspond to temporal segments, enabling "tasks" to be constructed automatically from the experience in the environment.

Meta-learning in the context of reinforcement learning has largely focused on model-free approaches (Duan et al., 2016; Wang et al., 2016; Sung et al., 2017; Al-Shedivat et al., 2017). However, these algorithms present even more (meta-)training sample complexity than non-meta model-free RL methods, which precludes them from real-world applications. Recent work (Sæmundsson et al., 2018) has developed a model-based meta RL algorithm, framing meta-learning as a hierarchical latent variable model, training for episodic adaptation to dynamics changes; the modeling is done with GPs, and results are shown on the cart-pole and double-pendulum agents. In contrast, we propose an approach for learning online adaptation of high-capacity neural network dynamics models; we present two instantiations of this general approach and show results on both simulated agents and a real legged robot.

## 3 PRELIMINARIES

In this section, we present model-based reinforcement learning, introduce the meta-learning formulation, and describe the two main meta-learning approaches.

### 3.1 MODEL-BASED REINFORCEMENT LEARNING

Reinforcement learning agents aim to perform actions that maximize some notion of cumulative reward. Concretely, consider a Markov decision process (MDP) defined by the tuple $(\mathcal{S}, \mathcal{A}, p, r, \gamma, \rho_0, H)$. Here, $\mathcal{S}$ is the set of states, $\mathcal{A}$ is the set of actions, $p(\mathbf{s}'|\mathbf{s}, \mathbf{a})$ is the state transition distribution, $r : \mathcal{S} \times \mathcal{A} \to \mathbb{R}$ is a bounded reward function, $\rho_0 : \mathcal{S} \to \mathbb{R}_+$ is the initial state distribution, $\gamma$ is the discount factor, and $H$ is the horizon. A trajectory segment is denoted by $\tau(i, j) := (\mathbf{s}_i, \mathbf{a}_i, ..., \mathbf{s}_j, \mathbf{a}_j, \mathbf{s}_{j+1})$. Finally, the sum of expected rewards from a trajectory is the return. In this framework, RL aims to find a policy $\pi : \mathcal{S} \to \mathcal{A}$ that prescribes the optimal action to take from each state in order to maximize the expected return.

Model-based RL aims to solve this problem by learning the transition distribution $p(\mathbf{s}'|\mathbf{s}, \mathbf{a})$, which is also referred to as the dynamics model. This can be done using a function approximator $\hat{p}_{\boldsymbol{\theta}}(\mathbf{s}'|\mathbf{s}, \mathbf{a})$ to approximate the dynamics, where the weights $\boldsymbol{\theta}$ are optimized to maximize the log-likelihood of the observed data $\mathcal{D}$. In practice, this model is then used in the process of action selection by either producing data points from which to train a policy, or by producing predictions and dynamics constraints to be optimized by a controller.

### 3.2 META-LEARNING

Meta-learning is concerned with automatically learning learning algorithms that are more efficient and effective than learning from scratch. These algorithms leverage data from *previous* tasks to acquire a learning procedure that can quickly adapt to new tasks. These methods operate under the assumption that the previous meta-training tasks and the new meta-test tasks are drawn from the same task distribution $\rho(\mathcal{T})$ and share a common structure that can be exploited for fast learning. In the supervised learning setting, we aim to learn a function $f_{\boldsymbol{\theta}}$ with parameters $\boldsymbol{\theta}$ that minimizes a supervised loss $\mathcal{L}_{\mathcal{T}}$. Then, the goal of meta-learning is to find a learning procedure, denoted as $\boldsymbol{\theta}' = u_{\boldsymbol{\psi}}(\mathcal{D}_{\mathcal{T}}^{\text{tr}}, \boldsymbol{\theta})$, that can learn a range of tasks $\mathcal{T}$ from small datasets $\mathcal{D}_{\mathcal{T}}^{\text{tr}}$.

We can formalize this meta-learning problem setting as optimizing for the parameters of the learning procedure $\boldsymbol{\theta}, \boldsymbol{\psi}$ as follows:

$$\min_{\boldsymbol{\theta}, \boldsymbol{\psi}} \; \mathbb{E}_{\mathcal{T} \sim \rho(\mathcal{T})} \left[ \mathcal{L}(\mathcal{D}_{\mathcal{T}}^{\text{test}}, \boldsymbol{\theta}') \right] \quad \text{s.t.} \quad \boldsymbol{\theta}' = u_{\boldsymbol{\psi}}(\mathcal{D}_{\mathcal{T}}^{\text{tr}}, \boldsymbol{\theta}) \tag{1}$$

where $\mathcal{D}_{\mathcal{T}}^{\text{tr}}, \mathcal{D}_{\mathcal{T}}^{\text{test}}$ are sampled without replacement from the meta-training dataset $\mathcal{D}_{\mathcal{T}}$.

Once meta-training optimizes for the parameters $\boldsymbol{\theta}_*, \boldsymbol{\psi}_*$, the learning procedure $u_{\boldsymbol{\psi}}(\cdot, \boldsymbol{\theta})$ can then be used to learn new held-out tasks from small amounts of data. We will also refer to the learning procedure $u$ as the update function.

**Gradient-based meta-learning.** Model-agnostic meta-learning (MAML) (Finn et al., 2017) aims to learn the initial parameters of a neural network such that taking one or several gradient descent steps from this initialization leads to effective generalization (or few-shot generalization) to new tasks. Then, when presented with new tasks, the model with the meta-learned initialization can be quickly fine-tuned using a few data points from the new tasks. Using the notation from before, MAML uses gradient descent as a learning algorithm:

$$u_{\boldsymbol{\psi}}(\mathcal{D}_{\mathcal{T}}^{\text{tr}}, \boldsymbol{\theta}) = \boldsymbol{\theta} - \alpha \nabla_{\boldsymbol{\theta}} \mathcal{L}(\mathcal{D}_{\mathcal{T}}^{\text{tr}}, \boldsymbol{\theta}) \tag{2}$$

The learning rate $\alpha$ may be a learnable paramter (in which case $\boldsymbol{\psi} = \alpha$) or fixed as a hyperparameter, leading to $\boldsymbol{\psi} = \varnothing$. Despite the update rule being fixed, a learned initialization of an overparameterized deep network followed by gradient descent is as expressive as update rules represented by deep recurrent networks (Finn and Levine, 2017).

**Recurrence-based meta-learning.** Another approach to meta-learning is to use recurrent models. In this case, the update function is always learned, and $\boldsymbol{\psi}$ corresponds to the weights of the recurrent model that update the hidden state. The parameters $\boldsymbol{\theta}$ of the prediction model correspond to the remainder of the weights of the recurrent model and the hidden state. Both gradient-based and recurrence-based meta-learning methods have been used for meta model-free RL (Finn et al., 2017; Duan et al., 2016). We will build upon these ideas to develop a meta model-based RL algorithm that enables adaptation in dynamic environments, in an online way.

## 4 META-LEARNING FOR ONLINE MODEL ADAPTATION

In this section, we present our approach for meta-learning for online model adaptation. As explained in Section 3.2, standard meta-learning formulations require the learned model $\boldsymbol{\theta}_*, \boldsymbol{\psi}_*$ to learn using $M$ data points from some new "task." In prior gradient-based and model-based meta-RL approaches (Finn et al., 2017; Sæmundsson et al., 2018), the $M$ has corresponded to $M$ trajectories, leading to episodic adaptation.

Our notion of task is slightly more fluid, where every segment of a trajectory can be considered to be a different "task," and observations from the past $M$ *timesteps* (rather than the past $M$ episodes) can be considered as providing information about the current task setting. Since changes in system dynamics, terrain details, or other environmental changes can occur at any time, we consider (at every time step) the problem of adapting the model using the $M$ past time steps to predict the next $K$ timesteps. In this setting, $M$ and $K$ are pre-specified hyperparameters; see appendix for a sensitivity analysis of these parameters.

In this work, we use the notion of environment $\mathcal{E}$ to denote different settings or configurations of a particular problem, ranging from malfunctions in the system's joints to the state of external disturbances. We assume a distribution of environments $\rho(\mathcal{E})$ that share some common structure, such as the same observation and action space, but may differ in their dynamics $p_{\mathcal{E}}(\mathbf{s}'|\mathbf{s}, \mathbf{a})$. We denote a trajectory segment by $\tau_{\mathcal{E}}(i, j)$, which represents a sequence of states and actions $(\mathbf{s}_i, \mathbf{a}_i, ..., \mathbf{s}_j, \mathbf{a}_j, \mathbf{s}_{j+1})$ sampled within an environment $\mathcal{E}$. Our algorithm assumes that the environment is locally consistent, in that every segment of length $j - i$ has the same environment. Even though this assumption is not always correct, it allows us to learn to adapt from data without knowing when the environment has changed. Due to the fast nature of our adaptation (less than a second), this assumption is seldom violated.

We pose the meta-RL problem in this setting as an optimization over $(\boldsymbol{\theta}, \boldsymbol{\psi})$ with respect to a maximum likelihood meta-objective. The meta-objective is the likelihood of the data under a predictive model $\hat{p}_{\boldsymbol{\theta}'}(\mathbf{s}'|\mathbf{s}, \mathbf{a})$ with parameters $\boldsymbol{\theta}'$, where $\boldsymbol{\theta}' = u_{\boldsymbol{\psi}}(\tau_{\mathcal{E}}(t - M, t - 1), \boldsymbol{\theta})$ corresponds to model parameters that were updated using the past $M$ data points. Concretely, this corresponds to the following optimization:

$$\min_{\boldsymbol{\theta}, \boldsymbol{\psi}} \ \mathbb{E}_{\tau_{\mathcal{E}}(t-M,t+K) \sim \mathcal{D}} \big[ \mathcal{L}(\tau_{\mathcal{E}}(t, t + K), \boldsymbol{\theta}'_{\mathcal{E}}) \big] \quad \text{s.t.:} \quad \boldsymbol{\theta}'_{\mathcal{E}} = u_{\boldsymbol{\psi}}(\tau_{\mathcal{E}}(t - M, t - 1), \boldsymbol{\theta}), \quad (3)$$

In that $\tau_{\mathcal{E}}(t - M, t + K) \sim \mathcal{D}$ corresponds to trajectory segments sampled from our previous experience, and the loss $\mathcal{L}$ corresponds to the negative log likelihood of the data under the model:

$$\mathcal{L}(\tau_{\mathcal{E}}(t, t + K), \boldsymbol{\theta}'_{\mathcal{E}}) \triangleq -\frac{1}{K} \sum_{k=t}^{t+K} \log \hat{p}_{\boldsymbol{\theta}'_{\mathcal{E}}}(\mathbf{s}_{k+1}|\mathbf{s}_k, \mathbf{a}_k). \quad (4)$$

In the meta-objective in Equation 3, note that the past $M$ points are used to adapt $\boldsymbol{\theta}$ into $\boldsymbol{\theta}'$, and the loss of this $\boldsymbol{\theta}'$ is evaluated on the future $K$ points. Thus, we use the past $M$ timesteps to provide insight into how to adapt our model to perform well for nearby future timesteps. As outlined in Algorithm 1, the update rule $u_{\boldsymbol{\psi}}$ for the inner update and a gradient step on $\boldsymbol{\theta}$ for the outer update allow us to optimize this meta-objective of adaptation. Thus, we achieve fast adaptation at test time by being able to fine-tune the model using just $M$ data points.

While we focus on reinforcement learning problems in our experiments, this meta-learning approach could be used for a learning to adapt online in a variety of sequence modeling domains. We present our algorithm using both a recurrence and a gradient-based meta-learner, as we discuss next.

**Gradient-Based Adaptive Learner (GrBAL).** GrBAL uses a gradient-based meta-learning to perform online adaptation; in particular, we use MAML (Finn et al., 2017). In this case, our update rule is prescribed by gradient descent ( 5.)

$$\boldsymbol{\theta}'_{\mathcal{E}} = u_{\boldsymbol{\psi}}(\tau_{\mathcal{E}}(t - M, t - 1), \boldsymbol{\theta}) = \boldsymbol{\theta}_{\mathcal{E}} + \boldsymbol{\psi} \nabla_{\boldsymbol{\theta}} \frac{1}{M} \sum_{m=t-M}^{t-1} \log \hat{p}_{\boldsymbol{\theta}_{\mathcal{E}}}(\mathbf{s}_{m+1}|\mathbf{s}_m, \mathbf{a}_m) \quad (5)$$

**Recurrence-Based Adaptive Learner (ReBAL).** ReBAL, instead, utilizes a recurrent model, which learns its own update rule (i.e., through its internal gating structure). In this case, $\boldsymbol{\psi}$ and $u_{\boldsymbol{\psi}}$ correspond to the weights of the recurrent model that update its hidden state.

**Algorithm 1** Model-Based Meta-Reinforcement Learning (train time)

---

**Require:** Distribution $\rho_{\mathcal{E}}$ over tasks
**Require:** Learning rate $\beta \in \mathbb{R}^+$
**Require:** Number of sampled tasks $N$, dataset $\mathcal{D}$
**Require:** Task sampling frequency $n_S \in \mathbb{Z}^+$
1: Randomly initialize $\boldsymbol{\theta}$
2: **for** $i = 1, \ldots$ **do**
3:    **if** $i \bmod n_S = 0$ **then**
4:       Sample $\mathcal{E} \sim \rho(\mathcal{E})$
5:       Collect $\tau_{\mathcal{E}}$ using Alg. 2
6:       $\mathcal{D} \leftarrow \mathcal{D} \cup \{\tau_{\mathcal{E}}\}$
7:    **end if**
8:    **for** $j = 1 \ldots N$ **do**
9:       $\tau_{\mathcal{E}}(t - M, t - 1), \tau_{\mathcal{E}}(t, t + K) \sim \mathcal{D}$
10:      $\boldsymbol{\theta}'_{\mathcal{E}} \leftarrow u_{\boldsymbol{\psi}}(\tau_{\mathcal{E}}(t - M, t - 1), \boldsymbol{\theta})$
11:      $\mathcal{L}_j \leftarrow \mathcal{L}(\tau_{\mathcal{E}}(t, t + K), \boldsymbol{\theta}'_{\mathcal{E}})$
12:    **end for**
13:    $\boldsymbol{\theta} \leftarrow \boldsymbol{\theta} - \beta \nabla_{\boldsymbol{\theta}} \frac{1}{N} \sum_{j=1}^{N} \mathcal{L}_j$
14:    $\boldsymbol{\psi} \leftarrow \boldsymbol{\psi} - \eta \nabla_{\boldsymbol{\psi}} \frac{1}{N} \sum_{j=1}^{N} \mathcal{L}_j$
15: **end for**
16: Return $(\boldsymbol{\theta}, \boldsymbol{\psi})$ as $(\boldsymbol{\theta}_*, \boldsymbol{\psi}_*)$

**Algorithm 2** Online Model Adaptation (test time)

---

**Require:** Meta-learned parameters $\boldsymbol{\theta}_*, \boldsymbol{\psi}_*$
**Require:** controller(), $H, r, n_A$
1: $\mathcal{D} \leftarrow \emptyset$
2: **for** each timestep $t$ **do**
3:    $\boldsymbol{\theta}'_* \leftarrow u_{\boldsymbol{\psi}_*}(\mathcal{D}(t - M, t - 1), \boldsymbol{\theta}_*)$
4:    $\mathbf{a} \leftarrow \text{controller}(\boldsymbol{\theta}'_*, r, H, n_A)$
5:    Execute $\mathbf{a}$, add result to $\mathcal{D}$
6: **end for**
7: Return rollout $\mathcal{D}$

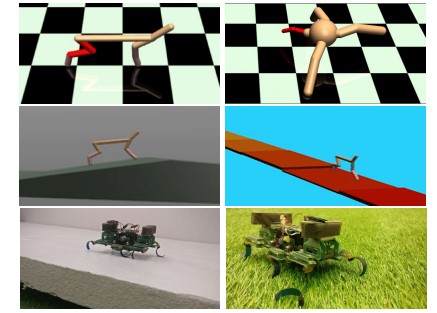

Figure 2: Two real-world and four simulated environments on which our method is evaluated and adaptation is crucial for success (e.g., adapting to different slopes and leg failures)

# 5   MODEL-BASED META-REINFORCEMENT LEARNING

Now that we have discussed our approach for enabling online adaptation, we next propose how to build upon this idea to develop a model-based meta-reinforcement learning algorithm. First, we explain how the agent can use the adapted model to perform a task, given parameters $\boldsymbol{\theta}_*$ and $\boldsymbol{\psi}_*$ from optimizing the meta-learning objective.

Given $\boldsymbol{\theta}_*$ and $\boldsymbol{\psi}_*$, we use the agent's recent experience to adapt the model parameters: $\boldsymbol{\theta}'_* = u_{\boldsymbol{\psi}_*}(\tau(t - M, t), \boldsymbol{\theta}_*)$. This results in a model $\hat{p}_{\boldsymbol{\theta}'_*}$ that better captures the local dynamics in the current setting, task, or environment. This adapted model is then passed to our controller, along with the reward function $r$ and a planning horizon $H$. We use a planning $H$ that is smaller than the adaptation horizon $K$, since the adapted model is only valid within the current context. We use model predictive path integral control (MPPI) (Williams et al., 2015), but, in principle, our model adaptation approach is agnostic to the model predictive control (MPC) method used.

The use of MPC compensates for model inaccuracies by preventing accumulating errors, since we replan at each time step using updated state information. MPC also allows for further benefits in this setting of online adaptation, because the model $\hat{p}_{\boldsymbol{\theta}'_{\mathcal{E}}}$ itself will also improve by the next time step. After taking each step, we append the resulting state transition onto our dataset, reset the model parameters back to $\boldsymbol{\theta}_*$, and repeat the entire planning process for each timestep. See Algorithm 2 for this adaptation procedure. Finally, in addition to test-time, we also perform this online adaptation procedure during the meta-training phase itself, to provide on-policy rollouts for meta-training. For the complete meta-RL algorithm, see Algorithm 1.

# 6   EXPERIMENTS

Our evaluation aims to answer the following questions: (1) Is adaptation actually changing the model? (2) Does our approach enable fast adaptation to varying dynamics, tasks, and environments, both inside and outside of the training distribution? (3) How does our method's performance compare to that of other methods? (4) How do GrBAL and ReBAL compare? (5) How does meta model-based RL compare to meta model-free RL in terms of sample efficiency and performance for these experiments? (6) Can our method learn to adapt online on a real robot, and if so, how does it perform? We next

present our set-up and results, motivated by these questions. Videos are available online[2], and further analysis is provided in the appendix. We first conduct a comparative evaluation of our algorithm, on a variety of simulated robots using the MuJoCo physics engine (Todorov et al., 2012). For all of our environments, we model the transition probabilities as Gaussian random variables with mean parameterized by a neural network model (3 hidden layers of 512 units each and ReLU activations) and fixed variance. In this case, maximum likelihood estimation corresponds to minimizing the mean squared error. We now describe the setup of our environments (Fig. 2), where each agent requires different types of adaptation to succeed at run-time:

**Half-cheetah (HC): disabled joint.** For each rollout during meta-training, we randomly sample a joint to be disabled (i.e., the agent cannot apply torques to that joint). At test time, we evaluate performance in two different situations: disabling a joint unseen during training, and switching between disabled joints during a rollout. The former examines extrapolation to out-of-distribution environments, and the latter tests fast adaptation to changing dynamics.

**HC: sloped terrain.** For each rollout during meta-training, we randomly select an upward or downward slope of low steepness. At test time, we evaluate performance on unseen settings including a gentle upward slope, a steep upward slope, and a steep hill that first goes up and then down.

**HC: pier.** In this experiment, the cheetah runs over a series of blocks that are floating on water. Each block moves up and down when stepped on, and the changes in the dynamics are rapidly changing due to each block having different damping and friction properties. The HC is meta-trained by varying these block properties, and tested on a specific (randomly-selected) configuration of properties.

**Ant: crippled leg.** For each meta-training rollout, we randomly sample a leg to cripple on this quadrupedal robot. This causes unexpected and drastic changes to the underlying dynamics. We evaluate this agent at test time by crippling a leg from outside of the training distribution, as well as transitioning within a rollout from normal operation to having a crippled leg.

In the following sections, we evaluate our model-based meta-RL methods (GrBAL and ReBAL) in comparison to several prior methods:

- **Model-free RL (TRPO)**: To evaluate the importance of adaptation, we compare to a model-free RL agent that is trained across environments $\mathcal{E} \sim \rho(\mathcal{E})$ using TRPO (Schulman et al., 2015).
- **Model-free meta-RL (MAML-RL)**: We compare to a state-of-the-art model-free meta-RL method, MAML-RL (Finn et al., 2017).
- **Model-based RL (MB)**: Similar to the model-free agent, we also compare to a single model-based RL agent, to evaluate the importance of adaptation. This model is trained using supervised model-error and iterative model bootstrapping.
- **Model-based RL with dynamic evaluation (MB+DE)**: We compare to an agent trained with model-based RL, as above. However, at test time, the model is adapted by taking a gradient step at each timestep using the past $M$ observations, akin to dynamic evaluation (Krause et al., 2017). This final comparison evaluates the benefit of explicitly training for adaptability.

All model-based approaches (MB, MB+DE, GrBAL, and ReBAL) use model bootstrapping, use the same neural network architecture, and use the same planner within experiments: MPPI (Williams et al., 2015) for the simulated experiments and random shooting (RS) (Nagabandi et al., 2017a) for the real-world experiments.

## 6.1 EFFECT OF ADAPTATION

First, we analyze the effect of the model adaptation, and show results from test-time runs on three environments: HC pier, HC sloped terrain with a steep up/down hill, and ant crippled leg with the chosen leg not seen as crippled during training. Figure 3 displays the distribution shift between the pre-update and post-update model prediction errors of three GrBAL runs, showing that using the past $M$ timesteps to update $\boldsymbol{\theta}_*$ (pre) into $\boldsymbol{\theta}'_*$ (post) does indeed reduce model error on predicting the following $K$ timesteps.

---

[2]Videos available at: https://sites.google.com/berkeley.edu/metaadaptivecontrol

## 6.2 Performance and Meta-training Sample Efficiency

We first study the sample efficiency of the meta-training process. Figure 4 shows the average return across test environments w.r.t. the amount of data used for meta-training. We (meta-)train the model-free methods (TRPO and MAML-RL) until convergence, using the equivalent of about two days of real-world experience. In contrast, we meta-train the model-based methods (including our approach) using the equivalent of 1.5-3 hours of real-world experience. Our methods result in superior or equivalent performance to the model-free agent that is trained with 1000 times more data. Our methods also surpass the performance of the non-meta-learned model-based ap-

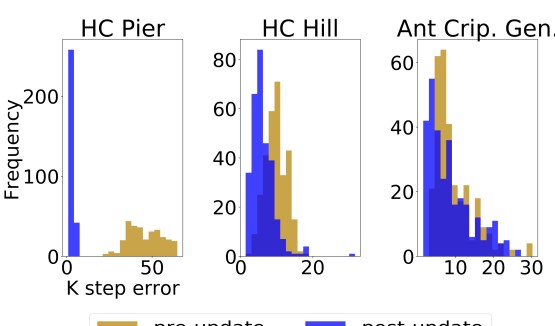

Figure 3: Histogram of normalized $K$-step model prediction errors of GrBAL, showing the improvement of the post-update model's predictions over the pre-update ones.

proaches. Finally, our performance closely matches the high asymptotic performance of the model-free meta-RL method for half-cheetah disabled, and achieves a suboptimal performance for ant crippled but, again, it does so with the equivalent of 1000 times less data. Note that this suboptimality in asymptotic performance is a known issue with model-based methods, and thus an interesting direction for future efforts. The improvement in sample efficiency from using model-based methods matches prior findings (Deisenroth and Rasmussen, 2011; Nagabandi et al., 2017a; Kurutach et al., 2018); the most important evaluation, which we discuss in more detail next, is the ability for our method to adapt online to drastic dynamics changes in only a handful of timesteps.

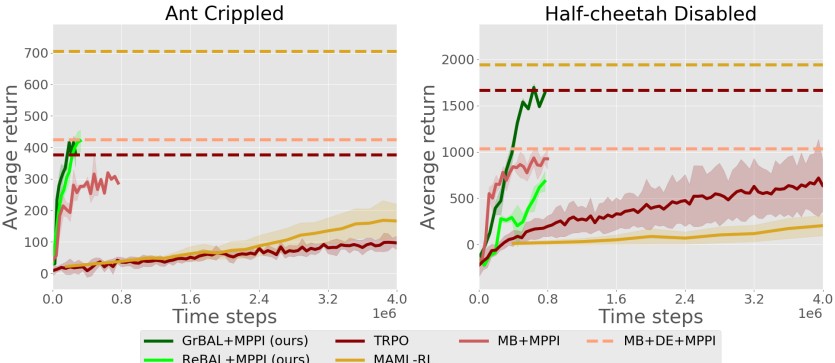

Figure 4: Compared to model-free RL, model-free meta-RL, and model-based RL methods, our model-based meta-RL methods achieve good performance with $1000\times$ less data. Dotted lines indicate performance at convergence. For MB+DE+MPPI, we perform dynamic evaluation at test time on the final MB+MPPI model.

## 6.3 Test-time Performance: Online Adaptation & Generalization

In our second comparative evaluation, we evaluate final test time performance both GrBAL and ReBAL in comparison to the aforementioned methods. In the interest of developing efficient algorithms for real-world applications, we operate all methods in the low data regime for all experiments: the amount of data available (meta-)training is fixed across methods, and roughly corresponds to 1.5-3 hours of real-world experience depending on the domain. We also provide the performance of a MB oracle, which is trained using unlimited data from only the given test environment (rather than needing to generalize to various training environments).

In these experiments, note that all agents were meta-trained on a distribution of tasks/environments (as detailed above), but we then evaluate their adaptation ability on unseen environments at test time. We test the ability of each approach to adapt to sudden changes in the environment, as well as to generalize beyond the training environments. We evaluate the fast adaptation (F.A.) component on the HC disabled joint, ant crippled leg, and the HC pier. On the first two, we cause a joint/leg of the robot to malfunction in the middle of a rollout. We evaluate the generalization component also on the

tasks of HC disabled joint and ant crippled leg, but this time, the leg/joint that malfunctions has not been seen as crippled during training. The last environment that we test generalization on is the HC slopped terrain for a hill, where the agent has to run up and down a steep slope, which is outside of the gentle slopes that it experienced during training. The results, shown in Fig. 5, show returns that are normalized such that the MB oracle achieves a return of

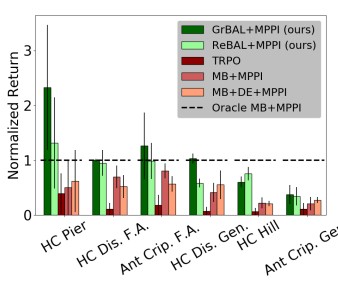

Figure 5: Simulated results in a variety of dynamic test environments. GrBAL outperforms other methods, even the MB oracle, in all experiments where fast adaptation is necessary. These results highlight the difficulty of training a global model, and the importance of adaptation.

In all experiments, due to low quantity of training data, TRPO performs poorly. Although MB+DE achieves better generalization than MB, the slow nature of its adaptation causes it to fall behind MB in the environments that require fast adaptation. On the other hand, our approach surpasses the other approaches in all of the experiments. In fact, in the HC pier and the fast adaptation of ant environments, our approach surpasses the model-based oracle. This result showcases the importance of adaptation in stochastic environments, where even a model trained with a lot of data cannot be robust to unexpected occurrences or disturbances. ReBAL displays its strengths on scenarios where longer sequential inputs allow it to better asses current environment settings, but overall, GrBAL seems to perform better for both generalization and fast adaptation.

## 6.4 Real-World Results

To test our meta model-based RL method's sample efficiency, as well as its ability to perform fast and effective online adaptation, we applied GrBAL to a real legged millirobot, comparing it to model-based RL (MB) and model-based RL with dynamic evaluation (MB+DE). Due to the cost of running real robot experiments, we chose the better performing method (i.e., GrBAL) to evaluate on the real robot. This small 6-legged robot, as shown in Fig. 1 and Fig. 2, presents a modeling and control challenge in the form of highly stochastic and dynamic movement. This robot is an excellent candidate for online adaptation for many reasons: the rapid manufacturing techniques and numerous custom-design steps used to construct this robot make it impossible to reproduce the same dynamics each time, its linkages and other body parts deteriorate over time, and it moves very quickly and dynamically with

The state space of the robot is a 24-dimensional vector, including center of mass positions and velocities, center of mass pose and angular velocities, back-EMF readings of motors, encoder readings of leg motor angles and velocities, and battery voltage. We define the action space to be velocity setpoints of the rotating legs. The action space has a dimension of two, since one motor on each side is coupled to all three of the legs on that side. All experiments are conducted in a motion capture room. Computation is done on an external computer, and the velocity setpoints are streamed over radio at 10 Hz to be executed by a PID controller on the microcontroller on-board of the robot.

We meta-train a dynamics model for this robot using the meta-objective described in Equation 3, and we train it to adapt on entirely real-world data from three different training terrains: carpet, styrofoam, and turf. We collect approximately 30 minutes of data from each of the three training terrains. This data was entirely collected using a random policy, in conjunction with a safety policy, whose sole purpose was to prevent the robot from exiting the area of interest.

Our first group of results (Table 1) show that, when data from a random policy is used to train a dynamics model, both a model trained with a standard supervised learning objective (MB) and a GrBAL model achieve comparable performance for executing desired trajectories on terrains from the training distribution.

Next, we test the performance of our method on what it is intended for: fast online adaptation of the learned model to enable successful execution of new, changing, or out-of-distribution environments at test time. Similar to the comparisons above, we compare GrBAL to a model-based method (MB) that involves neither meta-training nor online adaptation, as well as a dynamic evaluation method that involves online adaptation of that MB model (MB+DE). Our results (Fig. 6) demonstrate that GrBAL substantially outperforms MB and MB+DE, and, unlike MB and MB+DE, and that GrBAL can quickly 1) adapt online to a missing leg, 2) adjust to novel terrains and slopes, 3) account for miscalibration or errors in pose estimation, and 4) compensate for pulling payloads. None of

|           |       | Left | Str  | Z-z  | F-8  |
|-----------|-------|------|------|------|------|
| Carpet    | GrBAL | 4.07 | 3.26 | 7.08 | 5.28 |
|           | MB    | 3.94 | 3.26 | 6.56 | 5.21 |
| Styrofoam | GrBAL | 3.90 | 3.75 | 7.55 | 6.01 |
|           | MB    | 4.09 | 4.06 | 7.48 | 6.54 |
| Turf      | GrBAL | 1.99 | 1.65 | 2.79 | 3.40 |
|           | MB    | 1.87 | 1.69 | 3.52 | 2.61 |

Table 1: Trajectory following costs for real-world Gr-BAL and MB results when tested on three terrains that were seen during training. Tested here for left turn (Left), straight line (Str), zig-zag (Z-z), and figure-8 shapes (F-8). The methods perform comparably, indicating that online adaptation is not needed in the training terrains, but including it is not detrimental.

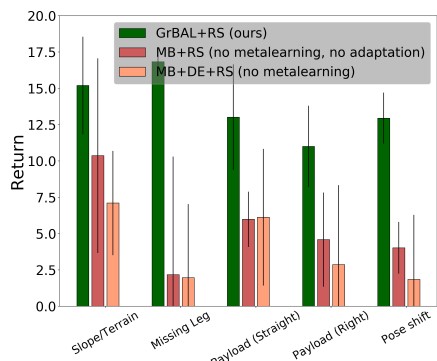

Figure 6: GrBAL clearly outperforms both MB and MB+DE, when tested on environments that (1) require online adaptation, and/or (2) were never seen during training.

these environments were seen during training time, but the agent's ability to learn how to learn enables it to quickly leverage its prior knowledge and fine-tune to adapt to new environments online. Furthermore, the poor performance of the MB and MB+DE baselines demonstrate not only the need for adaptation, but also the importance of good initial parameters to adapt from (in this case, meta-learned parameters). The qualitative results of these experiments in Fig. 7 show that the robot is able to use our method to adapt online and effectively follow the target trajectories, even in the presence of new environments and unexpected perturbations at test time.

bounding-style gaits; hence, its dynamics are strongly dependent on the terrain or environment at hand.

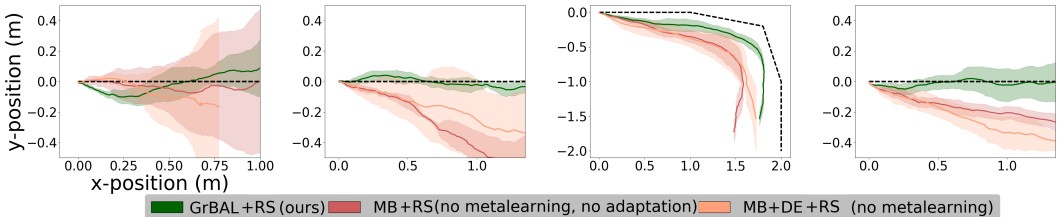

Figure 7: The dotted black line indicates the desired trajectory in the $xy$ plane. By effectively adapting online, our method prevents drift from a missing leg, prevents sliding sideways down a slope, accounts for pose miscalibration errors, and adjusts to pulling payloads (left to right). Note that none of these tasks/environments were seen during training time, and they require fast and effective online adaptation for success.

## 7 CONCLUSION

In this work, we present an approach for model-based meta-RL that enables fast, online adaptation of large and expressive models in dynamic environments. We show that meta-learning a model for online adaptation results in a method that is able to adapt to unseen situations or sudden and drastic changes in the environment, and is also sample efficient to train. We provide two instantiations of our approach (ReBAL and GrBAL), and we provide a comparison with other prior methods on a range of continuous control tasks. Finally, we show that (compared to model-free meta-RL approaches), our approach is practical for real-world applications, and that this capability to adapt quickly is particularly important under complex real-world dynamics.

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

## A    MODEL PREDICTION ERRORS: PRE-UPDATE VS. POST-UPDATE

In this section, we show the effect of adaptation in the case of GrBAL. In particular, we show the histogram of the $K$ step normalized error, as well as the per-timestep visualization of this error during a trajectory. Across all tasks and environments, the post-updated model $\hat{p}_{\boldsymbol{\theta}'_*}$ achieves lower prediction error than the pre-updted model $\hat{p}_{\boldsymbol{\theta}_*}$.

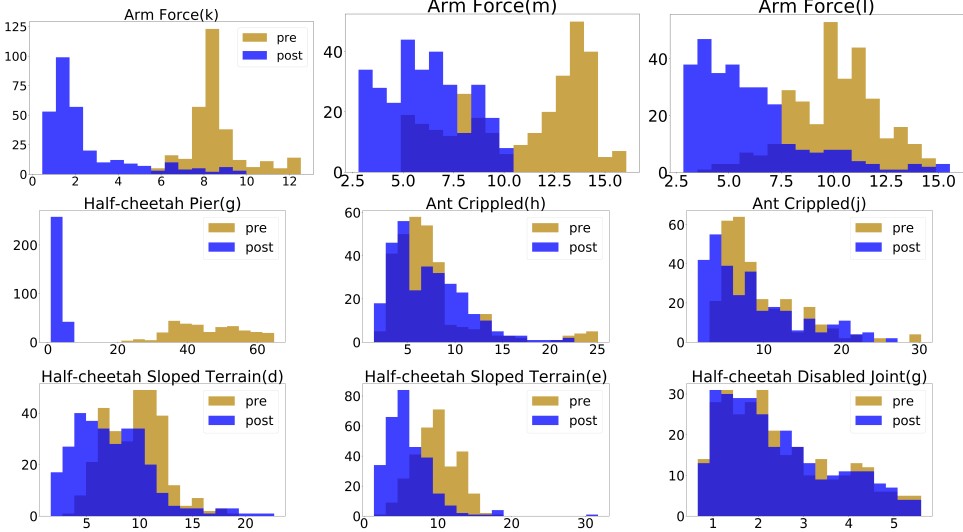

Figure 8: Histogram of the $K$ step normalized error across different tasks. GrBAL accomplishes lower model error when using the parameters given by the update rule.

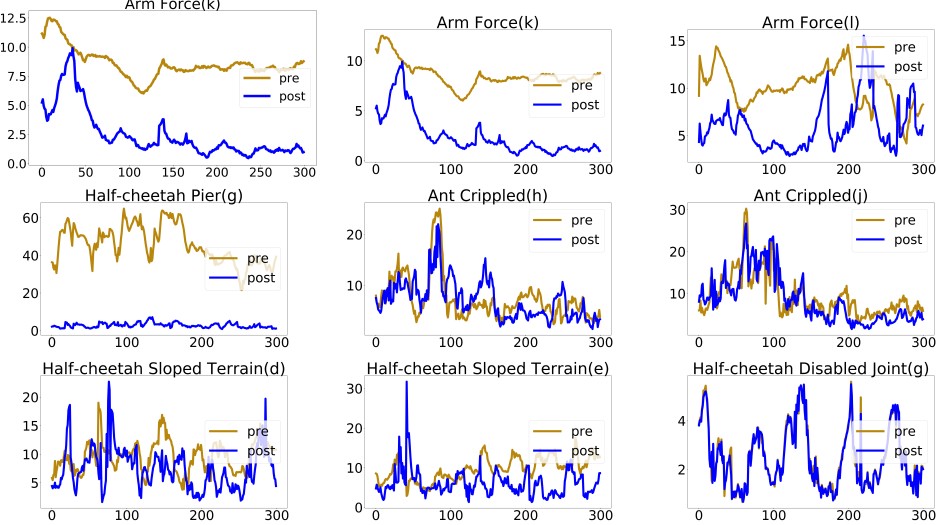

Figure 9: At each time-step we show the $K$ step normalized error across different tasks. GrBAL accomplishes lower model error using the parameters given by the update rule.

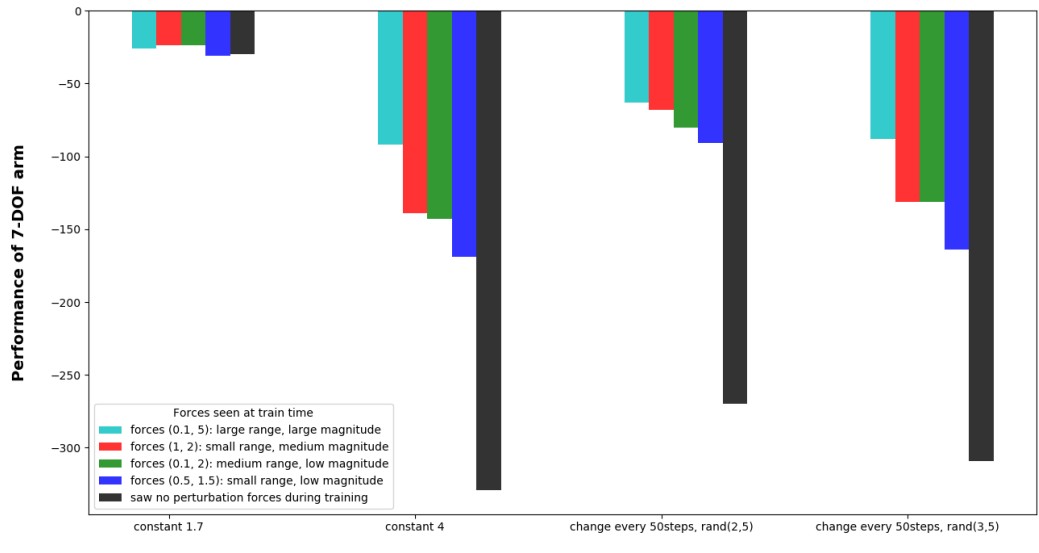

Figure 10: Effect of the meta-training distribution on test performance

## B  EFFECT OF META-TRAINING DISTRIBUTION

To see how training distribution affects test performance, we ran an experiment that used GrBAL to train models of the 7-DOF arm, where each model was trained on the same number of datapoints during meta-training, but those datapoints came from different ranges of force perturbations. We observe (in the plot below) that

1. Seeing more during training is helpful during testing — a model that saw a large range of force perturbations during training performed the best

2. A model that saw no perturbation forces during training did the worst

3. The middle 3 models show comparable performance in the "constant force = 4" case, which is an out-of-distribution task for those models. Thus, there is not actually a strong restriction on what needs to be seen during training in order for adaptation to occur at train time (though there is a general trend that more is better)

## C    SENSITIVITY OF K AND M

In this section we analyze how sensitive is our algorithm w.r.t the hyperparameters $K$ and $M$. In all experiments of the paper, we set $K$ equal to $M$. Figure 11 shows the average return of GrBAL across meta-training iterations of our algorithm for different values of $K = M$. The performance of the agent is largely unaffected for different values of these hyperparameters, suggesting that our algorithm is not particularly sensitive to these values. For different agents, the optimal value for these hyperparameters depends on various task details, such as the amount of information present in the state (a fully-informed state variable precludes the need for additional past timesteps) and the duration of a single timestep (a longer timestep duration makes it harder to predict more steps into the future).

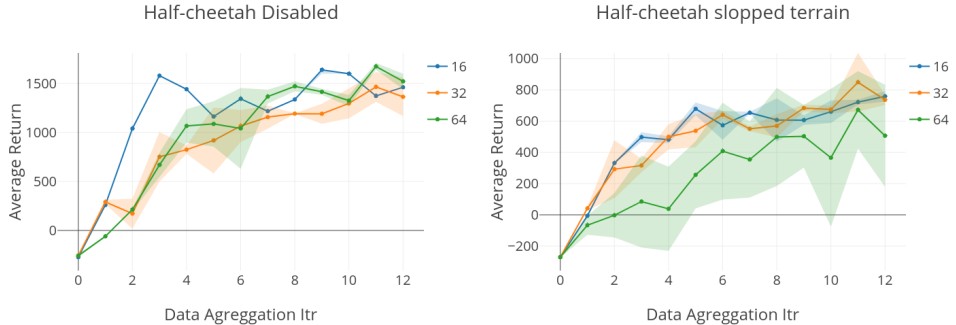

Figure 11: Learning curves, for different values of $K = M$, of GrBAL in the half-cheetah disabled and sloped terrain environments. The x-axis shows data aggreation iterations during meta-training, whereas the y-axis shows the average return achieved when running online adaptation with the meta-learned model from the particular iteration. The curves suggest that GrBAL performance is fairly robust to the values of these hyperparameters.

## D    REWARD FUNCTIONS

For each MuJoCo agent, the same reward function is used across its various tasks. Table 2 shows the reward functions used for each agent. We denote by $x_t$ the x-coordinate of the agent at time $t$, $\boldsymbol{ee}_t$ refers to the position of the end-effector of the 7-DoF arm, and $\boldsymbol{g}$ corresponds to the position of the desired goal.

Table 2: Reward functions

|  | Reward function |
|---|---|
| **Half-cheetah** | $\frac{x_{t+1}-x_t}{0.01} - 0.05\|\boldsymbol{a}_t\|_2^2$ |
| **Ant** | $\frac{x_{t+1}-x_t}{0.0e} - 0.005\|\boldsymbol{a}_t\|_2^2 + 0.05$ |
| **7-DoF Arm** | $-\|\boldsymbol{ee}_t - \boldsymbol{g}\|_2^2$ |

# E   HYPERPARAMETERS

Below, we list the hyperparameters of our experiments. In all experiments we used a single gradient step for the update rule of GrBAL. The learning rate (LR) of TRPO corresponds to the Kullback–Leibler divergence constraint. # Task/itr corresponds to the number of tasks sampled for collecting data to train the model or model, whereas # TS/itr is the total number of times steps collected (for all tasks). Finally, $T$ refers to the horizon of the task.

Table 3: Hyperparameters for the half-cheetah tasks

|  | LR | Inner LR | Epochs | K | M | Batch Size | # Tasks/itr | # TS/itr | T | $n_A$ Train | H Train | $n_A$ Test | H Test |
|---|---|---|---|---|---|---|---|---|---|---|---|---|---|
| **GrBAL** | 0.001 | 0.01 | 50 | 32 | 32 | 500 | 32 | 64000 | 1000 | 1000 | 10 | 2500 | 15 |
| **ReBAL** | 0.001 | - | 50 | 32 | 32 | 500 | 32 | 64000 | 1000 | 1000 | 10 | 2500 | 15 |
| **MB** | 0.001 | - | 50 | - | - | 500 | 64 | 64000 | 1000 | 1000 | 10 | 2500 | 15 |
| **TRPO** | 0.05 | - | - | - | - | 50000 | 50 | 50000 | 1000 | - | - | - | - |

Table 4: Hyperparameters for the ant tasks

|  | LR | Inner LR | Epochs | K | M | Batch Size | # Tasks/itr | # TS/itr | T | $n_A$ Train | H Train | $n_A$ Test | H Test |
|---|---|---|---|---|---|---|---|---|---|---|---|---|---|
| **GrBAL** | 0.001 | 0.001 | 50 | 10 | 16 | 500 | 32 | 24000 | 500 | 1000 | 15 | 1000 | 15 |
| **ReBAL** | 0.001 | - | 50 | 32 | 16 | 500 | 32 | 32000 | 500 | 1000 | 15 | 1000 | 15 |
| **MB** | 0.001 | - | 70 | - | - | 500 | 10 | 10000 | 500 | 1000 | 15 | 1000 | 15 |
| **TRPO** | 0.05 | - | - | - | - | 50000 | 50 | 50000 | 500 | - | - | - | - |

Table 5: Hyperparameters for the 7-DoF arm tasks

|  | LR | Inner LR | Epochs | K | M | Batch Size | # Tasks/itr | # TS/itr | T | $n_a$ Train | H Train | $n_a$ Test | H Test |
|---|---|---|---|---|---|---|---|---|---|---|---|---|---|
| **GrBAL** | 0.001 | 0.001 | 50 | 32 | 16 | 1500 | 32 | 24000 | 500 | 1000 | 15 | 1000 | 15 |
| **ReBAL** | 0.001 | - | 50 | 32 | 16 | 1500 | 32 | 24000 | 500 | 1000 | 15 | 1000 | 15 |
| **MB** | 0.001 | - | 70 | - | - | 10000 | 10 | 10000 | 500 | 1000 | 15 | 1000 | 15 |
| **TRPO** | 0.05 | - | - | - | - | 50000 | 50 | 50000 | 500 | - | - | - | - |

