# OpenReview forum: "Learning to Adapt in Dynamic, Real-World Environments through Meta-Reinforcement Learning"
_ICLR.cc/2019/Conference_

### Official Review · AnonReviewer3 · 2018-11-04
**This paper proposes a novel algorithm for online adaptation of a model-based RL approach, showing significant improvements in terms of speed, and also in terms of performance compared to standard approaches such as MAML-RL and non-adaptive model-based RL.**

**Rating:** 7
**Confidence:** 5

**Review:**

The authors introduce an algorithm that addresses the problem of online policy adaptation for model-based RL. The main novelty of the proposed approach is that it defines an effective algorithm that can easily and quickly adapt to the changing context/environments. It borrows the ideas from model-free RL (MAML) to define the gradient/recursive updates of their approach, and it incorporates it efficiently into their model-based RL framework. The paper is well written and the experimental results on synthetic and real world data show that the algorithm can quickly adapt its policy and achieve good results in the tasks, when compared to related approaches.

While applying the gradient based adaptation to the model-free RL is trivial and has  previously been proposed, in this work the authors do so by also focusing on the "local" context (M steps within a K-long horizon, allowing the method to  recover quickly if learning from contaminated data, and/or its global policy cannot generalize well to the local contexts. Although this extension is trivial it seems that it has not been applied and measured in terms of the adaptation "speed" in previous works. Theoretically, I see more value in their second approach where they investigate the application of fast parameter updates within model-based RL, showing that it does improve over the MAML-RL and non-adaptive model-based RL approaches. This is expected but  to my knowledge has not been investigated to this extent before.

What I find is lacking in this paper is insight into how sensitive the algorithm is in terms of the K/M ratio, and also how it affects the adaptation speed vs performance (tables 3-5 show an analysis but those are for different tasks); no theoretical analysis was performed to provide deeper understanding of it. The model does solve a practical problem (reducing the learning time and having more robust model), however, it would add more value to the current state of the art in RL if the authors proposed a method for optimal selection of the recovery points and also window ratio R/L depending on the target task. This would make a significant theoretical contribution and the method could be easily applicable to a variety of tasks. where the gains in the adaptation speed are important.

---

> ### Author Response · Authors · 2018-11-19
> **Thank you for your feedback!**
>
> We thank the reviewer for their valuable feedback and agree that the strength of our approach comes from being able to adapt the dynamics model to the local dynamics. We do include a model-free RL algorithm in our experiments, but this is a prior method that is included only for comparison: we clarify that both of our approaches are model-based, and neither are model-free.
>
> We also clarify that we do not choose "M steps within a K-long horizon." We have edited section 4 to paper to properly specify it. We use information from the past M steps to adapt the meta-learned model and predict the future K steps; this is done at every time-step of the rollout. In this setup, K and M are simply hyperparameters
>
> We have added to the appendix D a sensitivity analysis of the values K and M for GrBAL. The results show that our approach is not particularly sensitive to those values. We also added a discussion in Appendix D of how the values can be determined -- the optimal values depend on various task details, such as the amount of information present in the state (a fully-informed state variable precludes the need for additional past timesteps) and the duration of a single timestep (a longer timestep duration makes it harder to predict more steps into the future).
>
> Lastly, given our clarifications, it would really help us if the reviewer could clarify what they meant by "optimal selection of the recovery points" -- what does "recovery points" mean in this context?

---

### Official Review · AnonReviewer1 · 2018-11-06
**Important problem, minor technical contribution, missing related work and poor evaluation.**

**Rating:** 2
**Confidence:** 5

**Review:**

This work addresses the problem of online adapting dynamics models in the context of model-based RL. Learning globally accurate dynamics model is impossible if we consider that environments are dynamic and we can't observe every possible environment state at initial training time. Thus learning dynamics models that can be adapted online fast, to deal with unexpected und never seen before events is an important research problem.

This paper proposes to use meta-learning to train an update policy that can update the dynamics model at test time in a sample efficient manner. Two methods are proposed
- GrBAL: this method uses MAML for meta-learning
- ReBAL: this method trains a recurrent network during meta-training such that it can update the dynamics effectively at test time when the dynamics  change

Both methods are evaluated on several simulation environments, which show that GrBAL outperforms ReBAL (on average). GrBAL is then evaluated on a real system.

The strengths of this paper are:

- this work addresses an important problem and is well motivated
- experiments on both simulated and on a real system are performed

The weaknesses:

- the related work section is biased towards the ML community. There is a ton of work on adapting (inverse) dynamics models in the robotics community. This line of work is almost entirely ignored in this paper. Furthermore some important recent references for model-based RL are not provided in the related work section (PETS [3] and MPPI [2]), although MPPI is the controller that is used in this work as a framework for model-based RL. Additionally, existing work on model-based RL with meta-learning [1] has not been cited. This is unacceptable.
- There is no significant technical contribution - the "contribution" is that existing meta-learning methods have been applied to the model-based RL setting. Even if no-one has had that idea before - it would be a minor contribution, but given that there is prior work on meta-learning in the context of model-based RL, this idea itself is not novel anymore.
- Two methods are provided, without much analysis. Often authors refer to "our approach" - but it's actually not clear what they mean by our approach. The authors can't claim "model-based meta RL" as their approach.
- While I commend the authors for performing both simulation and real-world experiments, I find the that experiments lack a principled evaluation. More details below.

Feedback on experiments:

Section 6.2 (sample efficiency)

You compare apples to oranges here. I have no idea whether your improvements in terms of sample-efficiency are due to using a model-based RL approach or because your deploying meta-learning. It is well known that model-based RL is more sample efficient, but often cannot achieve the same asymptotic performance as model-free RL. Since MPPI is your choice of model-based RL framework, you would have to include an evaluation that shows results on MPPI with model bootstrapping (as presented in [2]) to give us an idea of how much more sample-efficient your approach is.

Section 6.3 (fast adaptation and generalization)

While in theory one can choose the meta-learning approach independently from the choice of model-based controller, in practice the choice of the MPC method is very important. MPPI can handle model inaccuracies very well - almost to the point where sometimes adaptation is not necessary. You CANNOT evaluate MPPI with online adaptation to another MPC approach with another model-learning approach. This does not give me any information of how your meta-learning improves model-adaptation. In essence these comparisons are meaningless. To make your results more meaningful you need to use the same controller setup (let's say MPPI) and then compare the following:
1. MPPI with your meta-trained online adaptation
2. MPPI results with a fixed learned dynamics model - this shows us whether online adaptation helps
3. results of MPPI with the initial dynamics model (trained in the meta-training phase) -without online adaptation. This will tell us whether the meta-training phase provides a dynamics model that generalizes better (even without online adaptation)
4. MPPI with model bootstrapping (as presented in [2]). This will show whether your meta-trained online adaptation actually outperforms simple online model bootstrapping in terms of sample-efficiency

The key here is that you need to use the same model-based control setup (whether its MPPI or some other method). Otherwise you cannot detangle the effect of controller choice from your meta-learned online adaptation.

6.4 Real-world: same comments as above, comparisons are not meaningful

[1] Meta Reinforcement Learning with Latent Variable Gaussian Processes, UAI 2018
[2] MPPI with model-bootstrapping: Information Theoretic MPC for Model-Based Reinforcement Learning , ICRA 2017
[3] Deep Reinforcement Learning in a Handful of Trials using Probabilistic Dynamics Models, NIPS 2018

---

> ### Author Response · Authors · 2018-11-09
> **Thank you for the feedback!**
>
> We thank the reviewer for the feedback.
>
> The main concern of the reviewer is that we did not control for the choice of controller. This is a misunderstanding. We implemented the same controller for all of the model-based comparisons; hence, all comparisons reported in the paper are fair. To be precise, we used MPPI for all simulation experiments, and random-shooting MPC for all real-world experiments (since the action spaces were of lower dimension and did not need iterations of refinement). We updated the paper to clarify this.
>
> Related work:
> We thank the reviewer for pointing out these recent works. We updated the paper to incorporate citations for model-based RL, adapting inverse dynamic models, and the suggested recent model-based RL citation.
>
> Sample efficiency:
> In this section, we do both of the things the reviewer mentions: we compare our MB meta-learning method against a state-of-the-art MF meta-learning method to show the benefit of model-based over model-free, and against a state-of-the-art MF method to show the benefit of meta-learning.
>
> Evaluation:
> The reviewer suggested 4 points to evaluate. Points (1) and (2) are exactly the results we show in Section 6.3: (1) corresponds to our full GrBAL/ReBAL, meta-learning with adaptation, and (2) corresponds to our MB baseline, which has neither meta-learning nor adaptation. We further have a DE baseline, which addresses the combination of adaptation without meta-learning.
>
> Point (3) suggests metalearning the prior with the adaptation objective, but then not adapting it at test-time. We ran this experiment on the real robot, and it performed worse than (1) and (2), failing to solve the task. This is expected; the meta-learned model parameters (theta*) were optimized to be used only after adapting them. We can add these numerical results to our results.
>
> The difference between the requested point (4) and our existing point (2) is the collection of expert data for initializing the training data set. Being able to collect expert samples is a strong assumption, requiring either human demonstrations or knowledge of the ground truth model, and does not fall under the assumptions of our problem setting.
>
> Contribution:
> Our contribution is a new model-based meta-RL algorithm that incorporates elements of meta-learning and model-based RL. While our method is relatively simple, we are not aware of prior works that show that meta-learning can be used to enable online adaptation to varying dynamics in the context of model-based RL. Further, our experiments, which include domains that are more complex than the cartpole and double pendulum in [1], demonstrate the effectiveness of the approach. If we are mistaken regarding prior works, please let us know!
>
> We would like to emphasize that our work presents an extensive comparative evaluation, and we believe that these results should be taken into consideration in evaluating our work. We compare multiple approaches across more than 6 simulated tasks as well as 4 tasks on a real-world robotic locomotion task. Hopefully our clarifications are convincing in terms of explaining why the evaluation is fair and rigorous, and we would of course be happy to modify it as needed.

---

> > ### Comment · AnonReviewer1 · 2018-11-15
> > **major concerns remain**
> >
> > Thank you for your response and addressing my concerns (at least partially). I'd like to re-iterate what my main concerns with this manuscript are. To summarize
> >
> > i ) work is not put in the context of existing relevant related work (not really addressed)
> > ii) minor/questionable technical contribution (not really addressed)
> > iii) evaluations are not designed to evaluate fast model adaptation (was partially addressed)
> >
> > In more detail:
> >
> > Before going into detail of my concerns, I'd like to quickly summarize your approach:
> >
> > 1. at train time you use a model-based RL algorithm to learn a dynamics model. You utilize existing meta-learning methods/ideas to learn representations that can be utilized to adapt the dynamics model fast at test time. Specifically you present a) GrBAL, at training time you use MAML to learn dynamics model parameters that can quickly be adapted to changes in the dynamics b) ReBAL, you learn a recurrent-based update policy that can update the dynamics model parameters effectively online.
> > 2. at test time you use a model-predictive controller with the learned dynamics model and adapt it online based on recent observations. At each controller time step you reset the dynamics model to the dynamics model learned in phase 1.
> >
> > In that context my concerns are:
> >
> > i) utilizing meta-learning in model-based RL is not a novel idea, yet you write most of your manuscript as if it were. Utilizing meta-learning to quickly adapt dynamics models online is also not novel, yet your writing makes readers believe that it is. While you've added the references I've mentioned, you have not really discussed how your proposed methods improve over other relevant work. Your introduction should highlight were current methods fall short, and how your proposed work improves over existing work.  Furthermore, you have not added any references for model-based control. There is a ton of related work that uses model-based controllers and adapts dynamics models online, with and without meta-learning. This needs to be acknowledged.
> >
> > ii) I'm still not clear on what your work exactly addresses. You're using 2 very different meta-learning approaches to learn models/update policies such that adaptation is fast at test time. Neither of them involve a significant contribution. Using MAML in this model-based RL context, reduces to using MAML in a regression problem (no technical advances here). Learning the recurrent-based update policy is something that has been extensively explored in the learning-to-learn community. It's not clear what you're adding here. You cite relevant work in the related work section, but you don't explain how your work differs from them. If there are technical issues that arise from applying these methods in the model-based RL framework, you do not describe them. Maybe there is a technical contribution here - but if there is you are not highlighting it.
> >
> > iii) when evaluating your methods, you want to highlight that your meta-learning approach leads to models/policies that adapt faster.  However, I can still not infer that this is true, here is why:
> >
> > 1. Section 6.2: It's not clear whether this evaluation evaluates sample efficiency at training (meta-learning) time or at test time (how many samples you need to adapt online). In either case, if you want to highlight sample efficiency of your proposed approach (meta-learning to learn models that adapt fast at run time), you need to compare to model-based RL methods that do not use your meta-learning approach. And you need to use the same model-predictive controller. There is no point in comparing to model-free methods here.
> >
> > 2. Section 6.3. You say you use the same model-predictive controller in your experiments for model-based RL (MPPI), however you cite other papers that do not use MPPI. For instance, you say  "a non- adaptive model-based method (“MB”), which employs a feedforward neural network as the dynamics model and selects actions using MPC (Nagabandi et al., 2017a), " your non-adaptive model-based method should be MPPI with a fixed  neural network model (ideally the same that you use to initialize your methods). This is particularly problematic, because recent model-based RL methods have by far outperformed the work you cite (Nagabandi et al., 2017a).   I want to re-iterate that you need to present the ablation study I suggested in my earlier review, and also present it as such (if you're already doing the experiments that I suggest, then change the plots and experiment description to make this clear) .
> >
> > to be cont'd

---

> > > ### Comment · AnonReviewer1 · 2018-11-15
> > > **continued**
> > >
> > > Some general comments:
> > > -----------------------
> > > The presentation of your approach would benefit from making it very explicit that you have a training time (model-based RL to learn models) and a run-time phase (model-predictive control with model adaptation).  This is a particularly confusing component of your evaluations, so please be clear about what phase your in, and if your evaluations are meant to evaluate run time adaptation then you should explain how all methods were initialized.
> > >
> > > Do not cite work as baseline, that you actually do not use as a baseline. MPPI with neural networks for dynamics models exists.
> > >
> > > to MPPI with model-bootstrapping you say:
> > >
> > > "The difference between the requested point (4) and our existing point (2) is the collection of expert data for initializing the training data set. Being able to collect expert samples is a strong assumption, requiring either human demonstrations or knowledge of the ground truth model, and does not fall under the assumptions of our problem setting."
> > >
> > >   I don't understand this comment. MPPI with model-bootstrapping does not require an initial training data set, but it would help of course. What I meant is that at run time, you could continue to update the model (so essentially you continue with the model-based RL setup) - the difference is that you're not resetting the model at each time step. You could argue that this is exactly not what you want to do, you don't want to update your model continuously. But then you should argue why you wouldn't want to do this, in your introduction.

---

> > > > ### Author Response · Authors · 2018-11-19
> > > > **Thanks again for the feedback**
> > > >
> > > > Thank you for taking the time to respond, we really appreciate your detailed feedback. We believe that we can address all of your concerns, please let us know if the revisions and modifications we describe below have addressed these issues. Thanks again for helping us improve the paper!
> > > >
> > > > (i) Meta-learning for online adaptation to dynamics has not been proposed in prior work. [1] trains for episodic adaptation, rather than online adaptation, showing good adaptation performance after several trials rather than several timesteps. We have clarified the introduction to scope the claims more carefully, but we do believe this is a novel contribution. However, if there is any other citation that covers this approach, we would be happy to reference it and discuss. We have made a best faith attempt to cover all topics you referenced in your comment.
> > > > Regarding prior work on model based meta-RL, to our knowledge [1] is the only prior work that uses both meta-learning and model-based RL together. If there are any others, we would be happy to cite and discuss them as well. While we agree that [1] makes a valuable contribution, the technique is very different from ours and is specific to non-parametric latent variable models, while our method addresses parametric models. Further, we explicitly train for online adaptation (i.e., using only M timesteps of data for adaptation). Instead, their approach trains for episodic adaptation (i.e. using around M trajectories). Finally, we evaluate our approach on a real-world robotic system, while the prior paper evaluates on cart-pole and pendulum.
> > > >
> > > > (ii) We have edited sections 2 and 4 to highlight the technical contributions over MAML. Our method is not a straightforward application of MAML to model-based RL. MAML requires a distribution over tasks to be hand-specified in advance. Our method removes this assumption by developing an online formulation of meta-learning where “tasks” correspond to segments of time and are provided implicitly by the environment. In addition with the empirical contributions, we believe that this does constitute a novel conceptual contribution.
> > > >
> > > > (iii)
> > > > - Regarding section 6.2 & MPPI: we added learning curves of MB and MB+DE to the plots. We edited all of the plot legends to clarify which planner is used for each method. All simulated comparisons use MPPI for all methods. We have fixed the citation of Nagabandi et. al. 2017a by replacing it with [2].
> > > >
> > > > - Regarding model bootstrapping: Sorry for the misunderstanding on our end. We edited the paper to clarify the following --- at training time, we iteratively collect and aggregate data using the MPPI with MPC for all model-based methods (our method, MB, and MB+DE), collecting data in the loop of training. As a result, our “MB” and “MB+DE” comparisons corresponds to MPPI with model-bootstrapping [2], with and without adaptation (respectively) when collecting roll-outs during training & run-time. For our method, we also use bootstrapping, meta-learning the dynamics models iteratively (see line 5 and 6 of Alg. 1). We therefore believe that our comparison is set up properly and that the paper adequately communicates this, but we would appreciate any feedback you might have here, and we would be happy to alter the comparison if needed.
> > > >
> > > > - “make it very explicit”: We edited the paper to make it clear that there is a training and run-time phase (which we refer to as meta-training and testing).
> > > >
> > > > Finally, we would emphasize that results on difficult problems, including substantial performance gains on 5 distinct tasks and including real-world robotic control problems, are also a contribution of our work. Algorithms that improve on prior work in terms of efficiency and generalization are of interest to the community, even when they build on ideas that were presented in prior work. If this were not the case, then most papers on model-based RL (a very old idea in itself) and RL for robotics would not be publishable. Therefore, we do not think that the criticism that there are other model-based RL papers, other meta-learning papers, or even other meta-learning model-based RL papers by itself precludes publication. We do however strongly agree with the reviewer that citing and discussing all relevant prior work, and appropriately scoping the claims, is critical, and we have endeavored to do so. We are grateful for any help and advice to do this better.
> > > >
> > > > [1] Meta Reinforcement Learning with Latent Variable Gaussian Processes, UAI 2018
> > > > [2] MPPI with model-bootstrapping: Information Theoretic MPC for Model-Based Reinforcement Learning , ICRA 2017

---

> > > > > ### Author Response · Authors · 2018-11-22
> > > > > **Regarding related work**
> > > > >
> > > > > Regarding prior work, as requested, we have extended the related work section by incorporating prior work on model-based control. In particular, we have added references on adaptive control methods [1-7], and online system identification [8]. Please let us know if we should include any specific paper, we will be happy to include it and discuss it.
> > > > >
> > > > >
> > > > > [1] Sastry, Sosale Shankara and Isidori, Alberto. Adaptive control of linearizable systems.IEEE Transactions on Automatic Control, 1989.
> > > > > [2] Meier, Franziska and Schaal, Stefan. Drifting Gaussian Processes with Varying Neighborhood Sizes for Online Model Learning. ICRA 2016.
> > > > > [3] Meier, Franziska and Kappler, Daniel and Ratliff, Nathan and Schaal, Stefan. Towards Robust Online Inverse Dynamics Learning. IROS 2016.
> > > > > [4] P. Pastor and Ludovic Righetti and M. Kalakrishnan and S Schaal. Online movement adaptation based on previous sensor experiences. IROS 2011.
> > > > > [5] Underwood, Samuel J and Husain, Iqbal. Online parameter estimation and adaptive control of permanent-magnet synchronous machines. Transactions on Industrial Electronics 2010
> > > > > [6] Kelouwani, Sousso and Adegnon, Kokou and Agbossou, Kodjo and Dube, Yves. Online system identification and adaptive control for PEM fuel cell maximum efficiency tracking. Transactions on Energy Conversion 2012.
> > > > > [7] Rai, Akshara and Sutanto, Giovanni and Schaal, Stefan and Meier, Franziska. Learning Feedback Terms for Reactive Planning and Control. ICRA 2017.
> > > > > [8] Manganiello, Patrizio and Ricco, Mattia and Petrone, Giovanni and Monmasson, Eric and Spagnuolo, Giovanni. Optimization of Perturbative PV MPPT Methods Through Online System Identification. Transactions on Industrial Electronics 2014.

---

### Official Review · AnonReviewer4 · 2018-11-12
**Shows superior sample complexity of model-based Meta RL, but not much further insight**

**Rating:** 7
**Confidence:** 3

**Review:**

The paper proposes using meta-learning and fast, online adaptation of models to overcome the mismatch between simulation and the real world, as well as unexpected changes and dynamics. This paper proposes two model-based meta-learning reinforcement algorithms, one based on MAML and the other based on recurrence, and experimentally shows how they are more sample efficient and faster at adapting to test scenarios than prior approaches, including prior model-free meta-learning approaches.

I do have an issue with the way this paper labels prior work as model-free meta-learning algorithms, since for example, MAML is a general algorithm that can be applied to model-free and model-based algorithms alike. It would be more accurate in my opinion to label the contributions of this paper as model-based instantiations of prior existing algorithms, rather than new algorithms outright.

I’m a bit confused with equation 3, as the expectation is over a single environment, and the trajectory of data is also sampled from a single environment. But in the writing, the paper describes the setting as a potentially different environment at every timestep. Equation 3 seems to assume that the  subsequence of data comes from a single environment, which contradicts what you say in the text. As described, equation 3 is then not really much different from previous episodic or task based formulations.

The results themselves are not unexpected, as there has already been prior work that this paper also mentions showing that model-based RL algorithms are more sample efficient than model-free.

Section 6.1, I like this comparison and showing how the errors are getting better.

For section 6.2, judging from the plots, it doesn’t seem you are doing any meta-learning in this experiment, so then are you just basically running a model-based RL algorithm? I’m very confused what you are trying to show. Are you trying to show the benefit of model-based vs model-free? Prior work has already done that. Are you trying to show that even just using a meta-learning algorithm in an online setting results in good online performance? Then you should be comparing your algorithm to just a model-based online RL algorithm. You also mention that the asymptotic performance falls behind, is this because your model capacity is low, or maybe your MPC method is insufficient? If so, then wouldn’t it be more compelling to, like prior work, combine this with a model-free algorithm and get the best of both worlds?

Section 6.3 results look good.

Section 6.4, I really like the fact you have results on a real robot.

Overall I think the paper does successfully show the sample complexity benefits and fast adaptation of model-based meta-RL methods. The inclusion of a real world robot experiment is a plus. However the result is not particularly surprising or insightful, as prior work has already shown the massive sample complexity improvement of model-based RL methods.

UPDATE (Dec 4, 2018):

I have read the author response and they have addressed the specific concerns I have brought up. I am overall positive about this paper and the new changes and additions so I will slightly increase my score, though I am still concerned about the significance of the results themselves.

---

> ### Author Response · Authors · 2018-11-19
> **Thank you for the feedback!**
>
> We thank the reviewer for the valuable feedback, and we clarify the individual points below. We have edited the paper to address each of the concerns raised in the review, and we would appreciate additional feedback regarding whether we have addressed the reviewer's concerns about the paper or if the reviewer has anything else they would like us to improve.
>
> "Results not unexpected":
> We agree that the sample efficiency of model-based RL is generally known, and we have revised Section 6.2 to explicitly state it. Our intent is not to claim that model-based RL is more sample-efficient than model-free RL (which, as the reviewer stated, is well known), but rather to show that meta-training for fast adaptation can improve over directly running online model updates with a model trained with standard model-based RL. Note that the comparison to "MB-DE" in Section 6.3 is precisely this comparison: adapting our meta-trained models outperforms adapting these standard model-based RL models by a large margin.
>
> The takeaway of this work is fast adaptation of expressive dynamics models. For instance, a real robot adapting online (in milliseconds) to unseen and drastic dynamics changes has not been shown in prior work that we know of. We emphasize that our meta-trained model can adapt in less than a second, whereas model-based RL from scratch takes minutes or hours.
>
> Relation to MAML/prior work:
> We have edited section 2 of the paper to clarify the relation to MAML. In summary: MAML assumes access to a hand-designed distribution of tasks. Instead, one of our primary contributions is the online formulation of meta-learning, where tasks correspond to temporal segments, enabling “tasks” to be constructed automatically from the experience in the environment; MAML is a very general algorithm, but it has not been previously demonstrated on online learning problems.
>
> Equation 3:
> We have fixed the discrepancy and added several clarifications in Section 4. Our method uses M consecutives steps to predict the next K steps, which makes the assumption that the environment is constant for M+K timesteps. As a result, only a fraction of the roll-out, i.e. M+K timesteps, has to correspond to the same environment. The underlying assumption is that the subsequence of data does indeed come from the same environment. In our experiments, M+K is 0.5 seconds, making this assumption true most of the time. The fast adaptation (F.A.) environments in Section 6.3 show this adaptation occurring as the environment keeps changing within the rollouts.
>
> Section 6.2:
> We fixed the typo that was originally in the caption: GrBAL and ReBAL are our proposed meta-learning algorithms, so there is indeed meta-learning in this experiment. In this plot, we aim to show that our model-based meta-learning approaches achieve high performance while using 1000x less data than the two model-free approaches. Finally, we edited this plot by adding two more comparisons to further clarify the benefit of our model-based meta-learning approach over standard non-meta-learned model-based approaches.
>
> The reviewer's comment about the asymptotic performance is very relevant, so we added it to the text in section 6.2. We agree that the development of some model-based/model-free hybrid would be great, and plan to do this in future work.

---

### Public Comment · (anonymous) · 2018-10-23
**The literature about adaptation/damage compensation with data-efficient RL is ignored**

I think this paper might be the first to apply meta-learning ideas to adaptation with a real robot. However, this is far from being the first paper to demonstrate that data-efficient reinforcement learning can be used for adapting to damage.

Unfortunately, the author of the submitted paper does not compare their result to this state-of-the-art... and does not even cite any of the previous paper on the topic (see below). This is worrisome because the previous papers require only 1-2 minutes of interaction time for adapting in similar tasks (legged robot with a blocked joint, a lost leg, etc.), compared to 1.5-3 hours in the submitted paper.

A few relevant papers about adaptation and damage recovery with data-efficient RL:

Active learning of a model / model-identification + direct policy search:
Bongard J, Zykov V, Lipson H. Resilient machines through continuous self-modeling. Science. 2006 Nov 17;314(5802):1118-21. http://www.cs.uvm.edu/~jbongard/papers/2006_Science_Bongard_Zykov_Lipson.pdf

Prior from simulation + Bayesian optimization:
Cully A, Clune J, Tarapore D, Mouret JB. Robots that can adapt like animals. Nature. 2015 May;521(7553):503. https://arxiv.org/pdf/1407.3501
See also: https://arxiv.org/pdf/1709.06919

Model-based policy search with priors:
Chatzilygeroudis K, Mouret JB. Using Parameterized Black-Box Priors to Scale Up Model-Based Policy Search for Robotics. 2018. Proc. of IEEE ICRA. https://arxiv.org/pdf/1709.06917

Repertoire of policies + high-level model:
Chatzilygeroudis K, Vassiliades V, Mouret JB. Reset-free trial-and-error learning for robot damage recovery. Robotics and Autonomous Systems. 2018 Feb 28;100:236-50. https://arxiv.org/abs/1610.04213

Bio-inspired approach:
Ren G, Chen W, Dasgupta S, Kolodziejski C, Wörgötter F, Manoonpong P. Multiple chaotic central pattern generators with learning for legged locomotion and malfunction compensation. Information Sciences. 2015 Feb 10;294:666-82. https://arxiv.org/abs/1407.3269

"Classic" RL:
Erden MS, Leblebicioğlu K. Free gait generation with reinforcement learning for a six-legged robot. Robotics and Autonomous Systems. 2008 Mar 31;56(3):199-212.

---

> ### Author Response · Authors · 2018-10-26
> **Thank you for your suggestions. Our work considers the problem of online adaptation (in less than a second) to various disturbances, rather than a trial-and-error method of adaptation (in minutes/hours) to damage.**
>
> Thank you for your suggestions. The biggest clarification that we would like to offer is that our method adapts online (in less than a second), and not minutes/hours. For example, when the agent sees a new terrain, when it encounters a slope, or when the system's pose estimation system become miscalibrated, we don't need to run multiple trials in this new setting (and we don't need an external reward signal like "distance travelled" to trigger/guide the adaptation). Instead, the agent constantly uses its past few data points, in a self-supervised way, to perform online adaptation of its dynamics model. Note that it successfully does this even when it encounters tasks that it did not see during training. This ability to adapt model parameters using such few data points is crucial, and we achieve it through meta-learning.
>
> We will add discussion of these works in the next version of our paper, to be thorough. We would, however, like to emphasize that the purpose of our work is not adapting to damage. The purpose of our work is an algorithm that uses meta-learning to enable *online* model adaptation. Although recovering from damage is included in our experiments, it is merely one example in this category of experiencing unexpected disturbances at test time: We also evaluate other tasks such as a pier of differing buoyancy from that seen during training, slopes that were never seen during training, and pulling an unknown payload. A big difference between our work and the suggested work is that we are not performing trial and error learning. The problem statement itself is very different, and thus it does not make sense to perform such a comparison.

---

### Author Response · Authors · 2018-11-26
**Paper updated to address reviewer feedback**

We believe that we addressed all of the reviewer's concerns. We would appreciate if the reviewers could take a look at our changes and let us know if they would like to revise their rating or request additional changes that would alleviate their concerns.

In summary, here are the main changes that we made to the paper:
- Ran a sensitivity analysis over the parameters K and M, and added a discussion section in the appendix regarding the selection of these values (R3)
- Edited the experiments section to clarify our main empirical insights (R4)
- Fixed the notational discrepancy in section 3 and added an explanation in section 4 regarding environments and rollouts (R4)
- Edited and added to the plot in section 6.2 to now include all experiments/comparisons of interest (R1, R4)
- Edited the related work to clarify the technical contributions of our method over MAML and prior work (R1, R4)
- Extended the related work section work to incorporate citations for model-based RL, adapting inverse dynamic models, and the suggested recent model-based RL citation (R1)
- Edited introduction to scope the claims more carefully (R1)
- Edited the experiments section's text and citations to clarify the misunderstanding regarding our choice of MPC controllers for each method of our comparisons (R1)
- Edited and clarified the methods and experiments to address all 4 of R1’s requested experimental comparisons (explicitly including #1/2/4, and running experiments to confirm that #3 does indeed fail) (R1)
- Edited the text to make it clear that we do already perform the suggested model-bootstrapping (R1)
- Edited the experiments to clearly differentiate meta-training time from test time (R1)

---

### Public Comment · (anonymous) · 2018-12-06
**Subjective assumption on whether the meta-test tasks are drawn from the same task distribution as the meta-training tasks**

This paper proposes a model-based meta-reinforcement learning method that achieves good results and enables fast adaptation in dynamics environment. While I can understand the paper better with the comments of the reviewers and the recent improvements done to the paper, there are still a few things that seem not so clear to me. It is without question that the assumption on whether the meta-test tasks are drawn from the same task distribution as the meta-training tasks is subjective , I would like to know how did the authors decide about the task distribution for training and testing? For example, for the experiments of half-cheetah (HC), the authors presented the results separately for HC disabled joint, HC slope terrains and HC pier and I suppose that three different models are trained for each of these experiments. Is this because of the differences in the task distributions between these three experiments? However, in the experiments with the millirobot, the authors meta-trained the agent on three different terrains with random trajectories, but tested the agent on various meta-test tasks such as missing leg, slope, added payload, etc that do not seem obvious to me that they are from the same task distribution as the meta-training tasks. Did the authors make assumption that these tasks are supposed to be in the same task distribution as the meta-training tasks? If so, why didn't the authors make the assumption that HC disabled joint, HC slope terrains and HC pier come from the same task distribution and just train one model for these three experiments, just like the experiments with the millirobot?

My second concern is that the deep neural networks architectures for the experiments are not at all mentioned. Since the expressive power of neural networks are limited by their size, so I wonder if the architecture and size of  deep neural networks will also limit the adaptation capability of the agent to different tasks.

Please correct me if I have some misunderstandings about the paper. Thank you.

If I am not mistaken, there is a typo on page 8, in section 6.3: "in comparison to the aforementioned methods."

---

> ### Author Response · Authors · 2018-12-06
> **Thank you for your comment**
>
> In the real-robot data collection is expensive, training a single model for the different terrains and conditions allows us to make a more efficient use of the data. Instead, in simulation we have separate experiments to have a more controlled comparison.
>
> The task distribution during training and testing does indeed need to match in theory. However, generating sufficiently diverse tasks requires a considerable engineering effort. It's often hard to get the kind of task diversity needed to evaluate things effectively with distributions that truly match. For instance, in the disabled half-cheetah task the agent just has 6 joints, so if we want one held-out joint there will be inevitably some distribution mismatch. While this is certainly a shortcoming in our experiments, we believe this setup overall is reasonable, and the comparison to prior methods is informative about overall performance of each method.
>
> Regarding the neural network size, we used a 3 layer NN with 512 units per layer and ReLU activations for all the feed-forward models, and a LSTM with 512 hidden units for the recurrent models. We will incorporate it to the paper.
>
> Please, let us know if you have further doubts or questions. And, thanks for pointing out the typo!

---

### Meta-Review · Area_Chair1 · 2018-12-14
**Promising work, should make sure final version carefully references robotics literature**

**Confidence:** 3
**Recommendation:** Accept (Poster)

**Metareview:**

The authors consider the use of MAML with model based RL and applied this to robotics tasks with very encouraging results. There was definite interest in the paper, but also some concerns over how the results were situated, particularly with respect to the related research in the robotics community. The authors are strongly encouraged to carefully consider this feedback, as they have been doing in their responses, and address this as well as possible in the final version.